# Curdepsidones B–G, Six Depsidones with Anti-Inflammatory Activities from the Marine-Derived Fungus *Curvularia* sp. IFB-Z10

**DOI:** 10.3390/md17050266

**Published:** 2019-05-05

**Authors:** Yi Ding, Faliang An, Xiaojing Zhu, Haiyuan Yu, Liling Hao, Yanhua Lu

**Affiliations:** State Key Laboratory of Bioreactor Engineering, East China University of Science and Technology, 130 Mei Long Road, Shanghai 200237, China; zjzsdy@163.com (Y.D.); flan2016@ecust.edu.cn (F.A.); 13761041863@163.com (X.Z.); y20160121@mail.ecust.edu.cn (H.Y.)

**Keywords:** marine-derived fungus, *Curvularia* sp. IFB-Z10, depsidones, ECD calculation, anti-inflammatory activity, *Propionibacterium acnes*

## Abstract

Six new depsidones, curdepsidones B–G (**1**–**6**), were obtained from the marine-derived fungus *Curvularia* sp. IFB-Z10. Their planar structures were determined by comprehensive analysis of HRESIMS and 1D/2D-NMR data. The absolute configuration of curdepsidones B–C (**1**–**2**) were established by synergistic use of DFT/NMR (density functional theory/nuclear magnetic resonance) and TDDFT/ECD (time-dependent density functional theory/electronic circular dichroism) calculations. Partial isolated compounds were tested for their anti-inflammatory activities in *Propionibacterium acnes*-induced THP-1 cells. Curdepsidone C (**2**) displayed significant anti-inflammatory properties with an IC_50_ value of 7.47 ± 0.35 μM, and reduced the *P. acnes*-induced phosphorylation levels of JNK and ERK in a dose-dependent mechanism. The possible anti-inflammatory mechanism of **2** was also investigated by molecular docking.

## 1. Introduction

In recent decades, marine-derived fungi have proven to be a promising and prolific source of natural products with intriguing carbon skeletons and/or active pharmacological properties [1,2,3]. Fungi from the marine environment always grow marine symbionts that coexist with animals as fish, sponge, algae, and soft coral in a relatively harsh environment, characterized by scarce nutrients and high osmotic pressure, and that produces many unimaginable biomedical potential secondary derivatives [4]. The strain *Curvularia* sp. IFB-Z10, isolated from the gut of a white croaker (*Argyrosomus argentatus*) collected from the Yellow Sea in China [5], is famous for its fascinating secondary metabolites, curvulamine, and curindolizine, with significant selective antibacterial and anti-inflammatory activities, respectively, when cultured in liquid media [6,7]. To further fertilize the chemical and biological diversity of its metabolites, *Curvularia* sp. IFB-Z10 was refermented in several different solid media rather than liquid media. Comparing the extract of various fermentation media by HPLC-DAD (high performance liquid chromatograph-diode array detector) profiles, the rice solid media was ultimately selected for large-scale cultivation and led to the isolation of several new polyketides [8,9] and depsidones [10] with significant cytotoxicity activities, which differ significantly from the liquid media in our previous study.

Depsidones basically occur as cyclic diaryl ethers with an ester linkage joining the two aromatic rings [11] and are mainly found in lichens [12,13,14], fungi [15,16], and plants [17,18], while rarely obtained from marine habitats. Most of them displayed antibacterial [12,19,20], antifungal [15,21], cytotoxicity [14,18,22], antimalarial [23], and immunomodulatory activities [24]. We had previously identified a new depsidone, curdepsidone A, which showed significant cytotoxicity against human hepatoma cell line BEL7402 and BEL7402/5-Fu with an IC_50_ of 9.85 and 2.46 μM, respectively [10]. To get more bioactive depsidones for further investigation from *Curvularia* sp. IFB-Z10 cultured in solid media, we scaled up the fungal fermentation in 300 flasks (1 L, 160 g rice, 210 mL 5% Ca^2+^ solution). Careful chemical investigation of the targeted extract, guided by the HPLC-DAD, resulted in the discovery of six new depsidones, curdepsidones B–G (**1**–**6**) (Figure 1). Elucidating the absolute structures of natural products with a flexible chain or complex ring system, available only in small quantities, is often exceptionally difficult, highlighted by a series of chemical structure revised reports, including baulamycin [25], cryptospirolepine [26], hypodoratoxide [27], myrtucommulone K [28], and duclauxins [29], etc. In this study, coupling constant analysis, DFT/NMR (density functional theory/nuclear magnetic resonance) and TDDFT/ECD (time-dependent density functional theory/electronic circular dichroism) calculations, was synergistically applied in the successful determination of absolute configuration of stereogenic centers C-8 and C-10 in curdepsidones B and C. Curdepsidone C (**2**) displayed significant anti-inflammatory properties by inhibitory effects against IL-1β release, with an IC_50_ value of 7.47 ± 0.35 μM in *Propionibacterium acnes*-induced THP-1 cells. Compound **2** could obviously suppress the *P. acnes*-induced phosphorylation of JNK and ERK in a dose-dependent manner. Herein, we described the isolation, structure elucidation, biological evaluation, and molecular docking of **2**.

## 2. Results and Discussion

Curdepsidone B (**1**) and curdepsidone C (**2**) were obtained as brown gum. Their molecular formulas were determined as C_22_H_24_O_10_ on the basis of HRESIMS (*m*/*z* 447.1295 [M − H]^−^, calcd. for [C_22_H_24_O_10_ − H]^−^, 447.1297) and ^13^C NMR data, requiring 11 degrees of unsaturation. The ^1^H NMR (Table 1) spectra of **1** exhibited one hydroxyl proton at δ_H_ 9.27 (s), two meta-substitution protons at δ_H_ 6.65 and 6.47 (d, *J* = 3.0 Hz), two oxygenated methine groups at δ_H_ 5.65 (dd, *J* = 11.1, 2.9 Hz) and 4.53 (dd, *J* = 10.5, 2.7 Hz), three methoxy groups at δ_H_ 3.79, 3.78, and 3.54 (s), two aromatic methyl groups at δ_H_ 2.46, 2.38 (s), and two diastereotopic methylene protons at δ_H_ 2.41 (m), 1.93 (ddd, *J* = 14.0, 10.5, 2.9 Hz). By the combination of ^13^C NMR (Table 2) and HSQC spectra, the 22 carbons were attributed to two benzene rings, two carbonyl carbons, three methoxy groups, two methyl groups, one methylene, and two methines. Comparing the NMR data with curdepsidone A [10], compounds **1** and **2** were inferred as depsidone analogues.

Further information about structure was derived from 2D NMR spectral analyses. The key HMBC correlations (Figure 2) from CH_3_-12 (δ_H_ 2.38, s) to C-1 (δ_C_ 113.6)/C-5 (δ_C_ 144.5)/C-6 (δ_C_ 137.2)/C-7 (δ_C_ 163.8), OH-4 (δ_H_ 9.27, s) to C-3 (δ_C_ 113.8)/C-4 (δ_C_ 154.2)/C-5, and H-8 (δ_H_ 5.65, dd, *J* = 11.1, 2.9 Hz) to C-3/C-4 suggested that a methyl group (δ_H_ 2.38) and a hydroxy group were located at C-6 and C-4 in the unit A of **1**, respectively. Meanwhile, the key HMBC correlations from OCH_3_-11 (δ_H_ 3.79, s) to C-11 (δ_C_ 175.4), H-10 (δ_H_ 4.53, dd, *J* = 10.5, 2.7 Hz ) to C-11/C-9 (δ_C_ 40.2)/C-8 (δ_C_ 75.9), H-9a (δ_H_ 1.93, ddd, *J* = 14.0, 10.5, 2.9 Hz) to C-10 (δ_C_ 67.3), H-9b (δ_H_ 2.41, m) to C-8, and OCH_3_-8 (δ_H_ 3.54, s) to C-8 and the key HMBC correlation from H-8 (δ_H_ 5.65, dd, *J* = 11.1, 2.9 Hz) to C-3 demonstrated the presence of a −CH(OCH_3_)-CH_2_-CH(OH)-COO-CH_3_ group, connected to C-3. In unit B, the two characteristic meta-substitution protons (δ_H_ 6.65, d, *J* = 3.0 Hz; 6.47,d, *J* = 3.0 Hz) in ^1^H NMR data and the HMBC correlations from CH_3_-7′ (δ_H_ 2.46, m) to C-1′ (δ_C_ 143.4)/C-5′ (δ_C_ 114.5)/C-6′ (δ_C_ 131.0) indicated that the unit B was a 1,2,3,5-tetrasubstituted benzene. However, the similar chemical shifts of oxygenated aromatic carbons lead to the equivocal assignment of the location of OCH_3_ (δ_H_ 3.54, s) and the planar structure of **1**. Three possible structures were speculated, as shown in Figure 3.

To solve the aforementioned problems, we took the ROESY spectra analysis and qccNMR method. Firstly, from the ROESY correlations of OCH_3_-5/CH_3_-12 (Appendix A), the OCH_3_ was identified to be located at C-5 rather than C-3’, suggesting that the structure of I was inconsiderable. Furthermore, the qccNMR method, which was useful in the determination of natural products without suitable crystals [30,31], was adopted in this study. The CLAD (corrected largest absolute deviation), CMAD (corrected mean absolute deviation), and *R*^2^ (the correlation coefficient) values of III (CLAD = 7.3, *R*^2^ = 0.9973) were more suitable than those of I (CLAD = 12.9, *R*^2^ = 0.9919) and II (CLAD = 8.0, *R*^2^ = 0.9951) (Appendix A). Thus, the OCH_3_ was located at C-5, and units A and B were connected by cyclic diaryl ethers with an ester linkage between C-2 and C-1′. The planar structures of **1** and **2** were determined by their similar 1D and 2D NMR data as shown. The obvious differences focused on the chemical shifts of H-8 and H-10 and their distinct ECD data (Figure 4) indicated that **1** and **2** were a couple of epimers.

As for the absolute configuration determination of **1** and **2**, the NMR chemical shift calculation and TDDFT/ECD calculations in chloroform solvent of four possible enantiomers succeeded in establishing the absolute configuration at C-8 and C-10 in **1** and **2**. The ECD calculation results showed that their exceptionally experimental ECD spectra were only useful in the absolute configuration determination of chiral center C-8 (Figure 4). The almost perfect match between the observed and computed ECD spectra showed the 8*R* and 8*S* in **1** and **2**, respectively. However, less cotton effects were influenced by the chiral center C-10, which was far away from chromophores. This challenge was solved by the synergistic application of coupling constant analysis, DFT/NMR and TDDFT/ECD calculations. The more suitable CMAD, CLAD, and *R*^2^ values in ^1^H NMR computed data assigned the *R* configuration to C-10 (Appendix A). Additionally, capitalizing on the empirical equations (the Karplus relationship for ^3^*J*_HH_ coupling constants) and the substitution patterns, which possessed a similar fragment to **1** [30,31], the absolute configuration of **1** was considered as 8*R*, 10*R* or 8*S*, 10*S* (Figure 5). Taking all of the above results in account, the absolute configurations of **1** and **2** were determined as 8*R*, 10*R* and 8*S*, 10*R*, respectively. Thus, the planar structures and absolute configurations of **1** and **2** were determined.

Curdepsidone D (**3**) and curdepsidone E (**4**) were obtained as brown gum. Their molecular formulas were determined as C_23_H_26_O_10_ on the basis of HRESIMS (*m*/*z* 461.1454 [M − H]^−^, calcd. for [C_23_H_26_O_10_ − H]^−^, 461.1453) and ^13^C NMR data, requiring 11 indices of hydrogen deficiency. The 1D NMR data of **3** and **4** were respectively similar to compound **1** and **2**, except for the replacement of the methoxy group in **1** with an ethoxy group (δ_H_ 3.73, q, *J* = 7.0 Hz and 1.31, t, *J* = 7.0 Hz; δ_C_ 66.7 and 15.3) in **3**. The final planar structures of **3** and **4** were established by detailed 2D NMR data analysis (Appendix A) as shown. And the *J* value of H-9/H-10 (^3^*J*_H9–H10_ 10.5 Hz) in **3** was similar to that of H-9/H-10 (^3^*J*_H9–H10_ 10.5 Hz) in **1**, while ^3^*J*_H9–H10_ = 5.5 Hz in **4** was similar to **2** (^3^*J*_H9–H10_ 5.5 Hz). In addition, the ECD data of **3** and **4** (Appendix A) were also in accordance with **1** and **2**, respectively. Thus, **3** and **4** were determined as analogues of compound **1** and **2**.

Curdepsidone F (**5**) was isolated as a pale yellow powder. Its molecular formula was determined as C_16_H_14_O_6_ on the basis of HRESIMS (*m*/*z* 301.0715 [M – H]^−^, calcd. for [C_16_H_14_O_6_ – H]^−^, 301.0718) and ^13^C NMR data, corresponding to 10 degrees of unsaturation. The ^1^H NMR (Table 1) data of **5** exhibited an isolated aromatic proton at δ_H_ 6.65 (s), two meta-substitution protons at δ_H_ 6.47 and 6.45 (d, *J* = 2.9 Hz), one methoxy group at δ_H_ 3.72 (s), and two aromatic methyl groups at δ_H_ 2.37, 2.35 (s). By the combination of ^13^C NMR (Table 2) and HSQC spectra, the 16 carbons were attributed to two benzene rings, one carbonyl carbon, one methoxy group, and two methyl groups, suggesting that **5** was a depsidone analogue. The ^1^H and ^13^C NMR data showed that **5** had an identical unit B to that of **1**. The difference was the absence of an aromatic methine carbon (C-3, δ_C_ 106.6) in unit A instead of a quaternary carbon, and the assignment was determined by the key HMBC correlations from H-3 (δ_H_ 6.65, s) to C-1 (δ_C_ 113.9)/C-2 (δ_C_ 160.6), H3-8 (δ_H_ 5.74, dd, *J* = 10.8, 3.5 Hz) to C-1/C-5 (δ_C_ 145.6)/C-6 (δ_C_ 137.6), and OCH_3_-5 (δ_H_ 3.72, s) to C-5. Accordingly, the final structure of compound **5** was determined as shown.

Curdepsidone G (**6**) was obtained as a pale yellow powder. The molecular formula was determined as C_18_H_18_O_7_ on the basis of HRESIMS (*m*/*z* 345.0977 [M – H]^−^, calcd. for [C_18_H_18_O_7_ – H]^−^, 345.0980) and ^13^C NMR data, indicating 10 degrees of unsaturation. The NMR spectra of **6** and **5** were highly similar. The main difference was that the aromatic methine (C-3, δ_C_ 106.6) in **5** was not found in **6**, while one quaternary carbon (δ_C_ 123.5), one methoxy group (δ_C_ 61.2), and one oxygenated methylene (δ_C_ 55.3) were presented in **6**. The assignment of these groups was confirmed by the key HMBC correlations from CH_2_OH-3 (δ_H_ 6.65, s) to C-2 (δ_C_ 156.1)/C-3 (δ_C_ 123.5)/C-4 (δ_C_ 155.8) and OCH_3_-4 (δ_H_ 4.01, s) to C-4. Thus, the structure of compound **6** was established as shown.

In addition, the plausible biosynthetic pathways involving condensation of orsellinic acid analogs and orcinol to form depsides, which then undergo oxidative coupling to yield depsidones, were included in Figure 6.

Depsidones **1**, **2**, **5**, and **6** were assayed for their anti-inflammatory activities by measuring the inhibitory effects of IL-1β production in *P. acnes*-induced THP-1 cells (Table 3) [32]. To avoid the false-positive consequences, the MTT assay was taken to test the cytotoxicities of these compounds, ensuring that the inhibition of IL-1β was not a result of the inhibition of cell proliferation. Compound **2** displayed notable inhibition on IL-1β production with an IC_50_ value of 7.47 ± 0.35 μM (retinoic acid as the positive control, IC_50_ 3.38 ± 0.28 μM), while **1** had no anti-inflammatory properties under its safe concentration, which revealed that the stereo-structure played a crucial role in the anti-inflammatory activities. Further research showed that curdepsidone C (**2**) could inhibit the production of IL-1β by selectively reducing the phosphorylation of JNK and ERK in a dose-dependent mechanism (Figure 7), and showed an unobvious effect on the phosphorylation of p38.

To investigate the possible action mechanism of compound **2**, we performed the molecular docking study on TLR2/1 protein receptor, which had a significant role in the innate immune response through recognizing microbial lipoproteins and lipopeptides [33]. As shown in Figure 8, hydrogen bonds and hydrophobic interactions were taken from **2** to the key residues Thr 297, Leu 299, Asn 327, and Lys 329. The results of molecular docking implied that the compound **2** may suppress the production of IL-1β through binding to the active site of the TLR2/1 protein.

## 3. Materials and Methods

### 3.1. General Experimental Procedures

Optical rotations were measured using a JASCO P-1020 polarimeter (JASCO Corporation, Tokyo, Japan) in MeOH at 25 °C. UV spectra were obtained with a Shimadzu UV-1800 spectrophotometer (Shimadzu Corporation, Tokyo, Japan). High-resolution electrospray ionization (HRESIMS) were recorded on an Agilent 6529B Q-TOF instrument (Agilent Technologies, Santa Clara, CA, USA). ECD spectra were carried out with Chirascan circular dichroism spectrometers (Applied Photophysics Ltd., Leatherhead, UK). Both 1D and 2D NMR spectra were measured with a Bruker AVⅢ-500 NMR spectrometer with tetramethylsilane (TMS) as an internal standard (Bruker, Karlsruhe, Germany). Preparative high-performance liquid chromatography (Pre-HPLC) was performed, utilizing a Shimadzu LC-20 system (Shimadzu, Tokyo, Japan), equipped with a Shim-pack RP-C_18_ column (20 × 250 mm i.d., 10 μm, Shimadzu, Tokyo, Japan) with a flow rate at 10 mL/min at 25 °C, recorded by a binary channel UV detector at 210 nm and 254 nm. Column chromatography (CC) was performed with silica gel (200–300 mesh, Qingdao Marine Chemical Inc., Qingdao, China) and ODS (50 μm, YMC, Kyoto, Japan). Thin-layer chromatography (TLC) was performed, using precoated silica gel GF254 plates (Qingdao Marine Chemical Inc., Qingdao, China).

### 3.2. Fungal Material

The strain *Curvularia* sp. IFB-Z10 was an endophytic fungus isolated from the gut of *Argyrosomus argentatus* captured from the Yellow Sea, which has been previously described [6].

### 3.3. Fermentation, Extraction, and Isolation

The fungus was incubated on potato dextrose agar (PDA) medium at 28 °C for approximately 5–6 days, then were cut into three agar pieces (nearly the size of 0.5 × 0.5 × 0.5 cm) and transferred into a 500 mL Erlenmeyer flask, containing 200 mL of potato dextrose broth (PDB). The flasks were cultured for 3 days at 28 °C on a rotary shaker at 180 rpm for inoculation. The seed cultures were added to the 300 × 1 L flasks containing rice medium (160 g rice, 210 mL 5% Ca^2+^ solution), previously sterilized at 121 °C for 25 min. All flasks were incubated at 28 °C for two weeks.

Following incubation, the solid rice cultures were extracted three times by EtOAc and the solvent was removed to dry under vacuum to give a crude extract (10.0 g). The extract was subjected to silica gel CC and fractionated with petroleum ether (PE), EtOAc, and MeOH, in order. The EtOAc fraction (3.0 g) was further separated on macroporous adsorbent resin with a stepped gradient elution with EtOH-H_2_O (30, 50, 75 and 100%). The 50% fraction was sequentially separated by an ODS column with MeOH-H_2_O (20% to 50%) to give five subfractions using the TLC. The subfraction B (280 mg) was loaded onto silica gel CC (CH_2_Cl_2_-MeOH, 30:1) and preparative HPLC (MeCN-H_2_O, 40:60, 10.0 mL/min) to yield compounds **1** (10.1 mg, *t*_R_ 31.1 min), **2** (6.4 mg, *t*_R_ 32.1 min), **5** (1.9 mg, *t*_R_ 38.1 min), and **6** (1.7 mg, *t*_R_ 33.9 min). Subfraction C (270 mg) was further purified by silica gel CC with a gradient of CH_2_Cl_2_-MeOH (100:1 to 50:1) and repeated preparative HPLC with MeOH-H_2_O (40:60, 10 mL/min) to give compounds **3** (4.4 mg, *t*_R_ 54.1 min) and **4** (1.7 mg, *t*_R_ 56.4 min).

### 3.4. ECD Measurement

ECD measurement were carried out with a Chirascan circular dichroism spectrometers at room temperature. All compounds were dissolved in CH_3_OH with an initial concentration of 1 mg/mL, and gradually diluted to their suitable concentration (**1**: 0.125 mg/mL; **2**: 0.125 mg/mL; **3**: 0.25 mg/mL, **4**: 0.25 mg/mL; **5**: 0.125 mg/mL; **6**: 0.125 mg/mL).

### 3.5. Computational Methods

#### 3.5.1. Conformational Analysis

The conformational analysis was primarily carried out by MOE 2014 with a LowMode MD algorithm at MMFF94x force field, with an RMSD threshold of 0.25 Å and energy window of 5 kcal/mol for undetermined relative configurations **a** and **b** of possible structures **Ⅰ**, **Ⅱ** and **III**. (Appendix A, enantiomers not shown). The energies of total conformers were listed in Appendix A.

#### 3.5.2. NMR Calculation

The theoretical calculations were performed by Gaussian 09. Firstly, all conformers were consecutively optimized at PM6 and HF/6-31G (d) theory levels. On the basis of Boltzmann distribution law (Equation (1)), dominant conformers were saved with their values of room-temperature equilibrium populations over 1%. These conformers were ultimately optimized at B3LYP/6-311G (d, p) in the gaseous phase, and the stability of structures was confirmed by vibrational frequency analysis (Appendix A). NMR calculations were performed by following the protocol adapted from Michael et al. [34] with the Gauge-Including Atomic Orbitals (GIAO) method at the mPW1PW91/6-311+G (2d, p) level in methanol, simulated by the IEFPCM model (Appendix A). In the end, the average values of the TMS-corrected NMR chemical shift were obtained from the Boltzmann distribution and the experimental values fitted by linear regression. The calculated ^13^C- and ^1^H-NMR chemical shift values of TMS in chloroform were 187.18 and 31.73 ppm.
(1)$$\frac{N_{i}}{N}=\frac{g_{i}e^{-\frac{E_{i}}{k_{B}T}}}{\sum g_{i}e^{-\frac{E_{i}}{k_{B}T}}}$$
where *N_i_* is the number of conformers *i* with energy *E_i_* and degeneracy *g_i_* at temperature *T* and *k*_B_ is Boltzmann constant.

#### 3.5.3. ECD calculation

The configurations of **1** and **2** for ECD calculations were obtained from former optimizations. The calculation was carried out in methanol, using the IEFPCM model by TD-DFT, and the 30 excited states of rotatory strengths were calculated. The ECD spectrum was performed in SpecDis [35], based on overlapping Gaussian functions for each transition, according to Equation (2):(2)$$\Delta \varepsilon (E)=\frac{1}{2.297\times 10^{-39}}\times \frac{1}{\sqrt{2\pi \sigma }}\sum_{i}^{A}\Delta E_{i}R_{i}e^{-{\left(\frac{E-E_{i}}{2\sigma }\right)}^{2}}$$
where *σ* represents the width of the band at 1/*e* height and Δ*E_i_* and *R_i_* are the excitation energies and rotatory strengths for transition *i*, respectively. Parameters of *σ* and UV-shift values were 0.49 eV and 7 nm for configuration **1** and 0.47 eV and 6 nm for configuration **2**, respectively. The spectra of enantiomers were produced directly by mirror inversions.

Curdepsidone B (**1**): Brown gum; [*α*]${}_{D}^{25}$ +92.0 (*c* 0.1, MeOH); UV (MeOH) *λ*_max_ (log*ε*) 207 (5.47), 265 (4.87) nm; ^1^H and ^13^C NMR (CDCl_3_), see Table 1 and Table 2; MS (ESI): *m*/*z* = 447.2 [M − H]^−^; negative HR-ESI-MS *m*/*z* 447.1295 [M − H]^−^, (calcd. for [C_22_H_24_O_10_ − H]^−^, 447.1297).

Curdepsidone C (**2**): Brown gum; [*α*]${}_{D}^{25}$ –124.0 (*c* 0.1, MeOH); UV (MeOH) *λ*_max_ (log*ε*) 207 (5.33), 269 (4.68) nm; ^1^H and ^13^C NMR (CDCl_3_), see Table 1 and Table 2; MS (ESI): *m*/*z* = 447.2 [M − H]^−^; negative HR-ESI-MS *m*/*z* 447.1295 [M − H]^−^, (calcd. for [C_22_H_24_O_10_ − H]^−^, 447.1297).

Curdepsidone D (**3**): Brown gum; [*α*]${}_{D}^{25}$ +110.0 (*c* 0.07, MeOH); UV (MeOH) *λ*_max_ (log*ε*) 207 (5.37), 268 (4.71) nm; ^1^H and ^13^C NMR (CDCl_3_), see Table 1 and Table 2; MS (ESI): *m*/*z* = 461.2 [M − H]^−^, 485.4 [M + Na]^+^; negative HR-ESI-MS *m*/*z* 461.1454 [M − H]^−^, (calcd. for [C_23_H_26_O_10_ − H]^−^, 461.1453).

Curdepsidone E (**4**): Brown gum; [*α*]${}_{D}^{25}$ –145.7 (*c* 0.1, MeOH); UV (MeOH) *λ*_max_ (log*ε*) 207 (5.43), 268 (4.78) nm; ^1^H and ^13^C NMR (CDCl_3_), see Table 1 and Table 2; MS (ESI): *m*/*z* = 461.3 [M − H]^−^, 485.5 [M + Na]^+^; negative HR-ESI-MS *m*/*z* 461.1453 [M − H]^−^, (calcd. for [C_23_H_26_O_10_ − H]^−^, 461.1453).

Curdepsidone F (**5**): Pale yellow powder; [*α*]${}_{D}^{25}$ –4.4 (*c* 0.09, MeOH); UV (MeOH) *λ*_max_ (log*ε*) 207 (5.66), 266 (5.12) nm; ^1^H and ^13^C NMR (CD_3_OD), see Table 1 and Table 2; MS (ESI): *m*/*z* = 301.1 [M − H]^−^, 303.3 [M + H]^+^; negative HR-ESI-MS *m*/*z* 301.0715 [M − H]^−^, (calcd. for [C_16_H_14_O_6_ − H]^−^, 301.0718).

Curdepsidone G (**6**): Pale yellow powder; [*α*]${}_{D}^{25}$ –12.0 (*c* 0.1, MeOH); UV (MeOH) *λ*_max_ (log*ε*) 207 (5.45), 258 (4.73) nm; ^1^H and ^13^C NMR (CDCl_3_), see Table 1 and Table 2; MS (ESI): *m*/*z* = 345.2 [M − H]^−^; negative HR-ESI-MS *m*/*z* 345.0977 [M − H]^−^, (calcd. for [C_18_H_18_O_7_ − H]^−^, 345.0980).

### 3.6. Anti-Inflammatory Activity Assay

#### 3.6.1. Cell Culture and Cell Viability Assay

The human monocytic cell line, THP-1 (Cell Bank of China Science Academy, Shanghai, China) and *P. acnes* (ATCC6919, Xiangfu biotech, Shanghai, China), were used for the anti-inflammatory assay. THP-1 cells were cultured in RPMI1640 medium with 10% fetal bovine serum (FBS, Gibco, NY, USA) in a humidified incubator (37 °C, 5% CO_2_). *P. acnes* bacteria were incubated in Cooked Meat Medium, containing cooked beef granules (Rishui biotechnology, Qingdao, China) in an anaerobic environment. The THP-1 cells were stimulated by the *P. acnes*, harvested at the exponential phase. The viability of THP-1 cells was evaluated by the MTT assay, specifically, seeding the THP-1 cells in 96-well plates at a density of 2 × 10^5^ cells/well and treated with serially diluted compounds for 36 h (37 °C, 5% CO_2_). After that, adding 20 µL MTT regent (5 mg/mL, Genetimes Technology Inc., Shanghai, China) to each well and incubating the samples at 37 °C for 4 h. Removing the supernatant, the formazan crystals were fully solubilized in DMSO (150 µL), and the absorbance was measured at 570 nm and 630 nm.

#### 3.6.2. Enzyme-Linked Immunosorbent Assay

The THP-1 cells were cultured in 96-well plates (2 × 10^5^ cells/well) in RPMI1640 medium without FBS, treated with different concentrations of compounds and incubation for 4 h. Subsequently, the live *P. acnes* bacteria were added to stimulate the cells for 24 h, and the cell culture supernatants were collected to analyze the levels of IL-1β with ELISA assays (Genetimes Technology Inc., Shanghai, China). Retinoic acid was used as the positive control [32,36].

#### 3.6.3. Molecular Docking Study

Molecular docking was carried out by AutoDock v4.2 software (The Scripps Research Institute, La Jolla, CA, USA). The 2D structure of compound **2** was converted into a 3D pdb format, using the Chem3D 16.0 software (CambridgeSoft Corporation, Cambridge, MA, USA). The crystal structure of protein receptor TLR2/1 was taken from the Protein Data Bank (PDB ID: 2Z7X) [33]. The whole protein was selected as the active site, and the interaction between the target protein and ligand was run with 250 individuals as described previously [37]. Finally, the interaction possessing the lowest energy was selected as the most dependable conformation.

#### 3.6.4. Statistical Analysis 

Statistical analysis was carried out using SPSS version 19 (IBM SPSS, Armonk, NY, USA). The data were presented as the mean ± standard deviation. One-way analysis of variance (ANOVA) was performed when the data involved three or more groups. *p* < 0.05 was considered as statistical significance.

## 4. Conclusions

In conclusion, a total of six new depsidone derivatives (**1**–**6**) were isolated from the marine-derived fungus *Curvularia* sp. IFB-Z10. Their structures were elucidated by synergistic application of coupling constant analysis, DFT/NMR and TDDFT/ECD calculations. The anti-inflammatory activities of compounds **1**, **2**, **5**, and **6** were evaluated wherein compound **2** showed notable inhibition on IL-1β production with an IC_50_ value of 7.47 ± 0.35 μM in *P. acnes*-induced THP-1 cells, and reduced the phosphorylation of JNK and ERK in a dose-dependent manner. Our research enriches the structure and activity diversity of the marine-derived natural depsidones.

## Figures and Tables

**Figure 1 marinedrugs-17-00266-f001:**
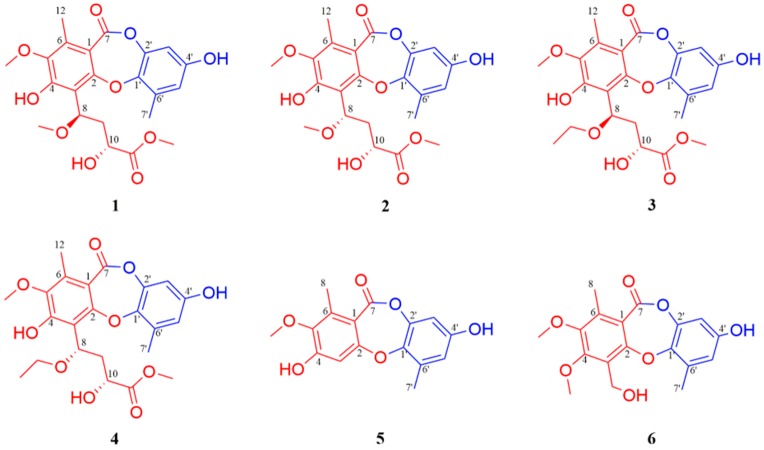
The structures of compounds **1**–**6**.

**Figure 2 marinedrugs-17-00266-f002:**
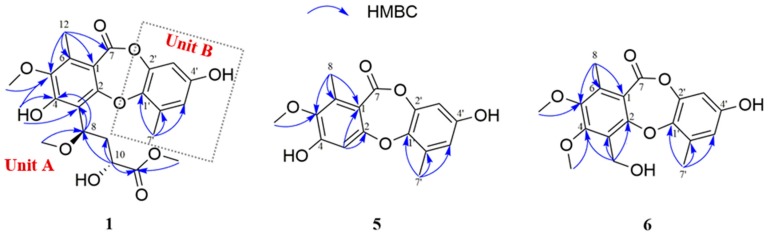
The key HMBC correlations of compounds **1**, **5**, and **6**.

**Figure 3 marinedrugs-17-00266-f003:**
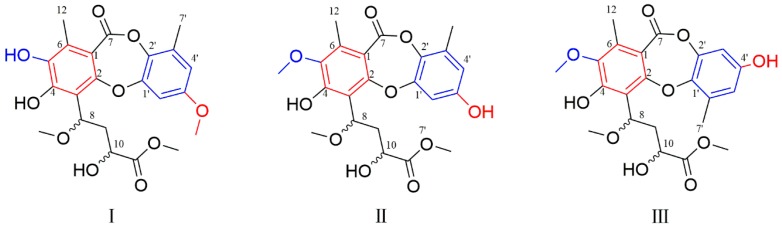
The possible structures of **1**.

**Figure 4 marinedrugs-17-00266-f004:**
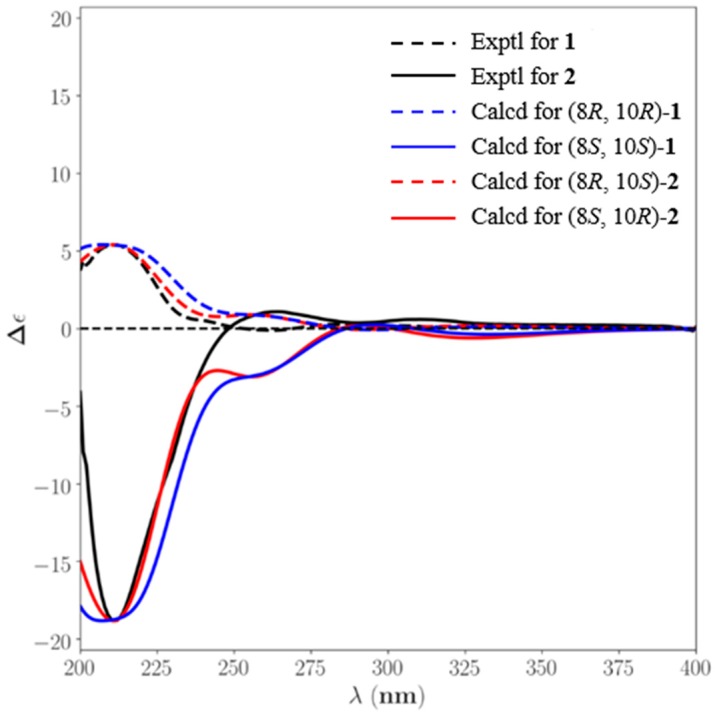
Calculated electronic circular dichroism (ECD) spectra of compound **1**, **2** and their enantiomers were compared with the experimental.

**Figure 5 marinedrugs-17-00266-f005:**
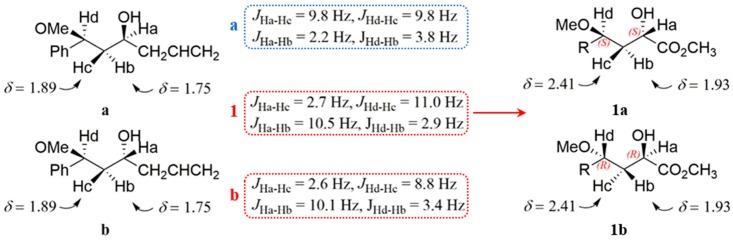
The possible absolute configuration of **1**. The compounds **a** and **b** were substitution patterns from the literature. The measured ^3^*J*_HH_ with the C8–C10 segment in **1** was similar to **b**.

**Figure 6 marinedrugs-17-00266-f006:**
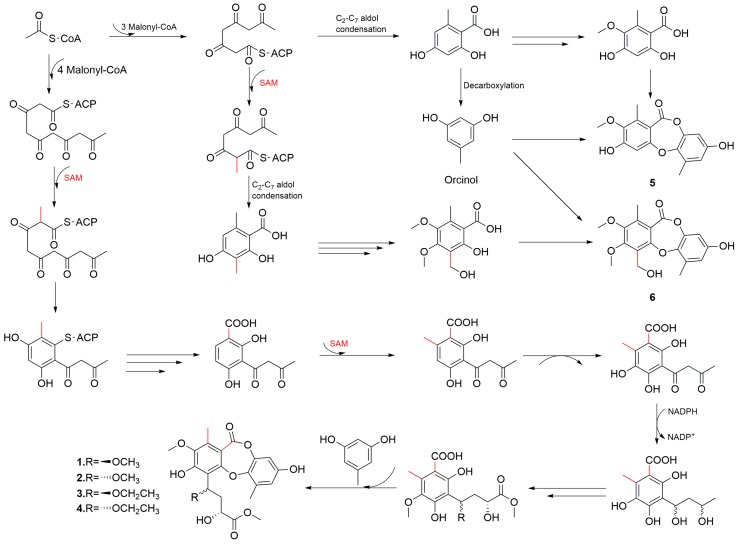
Plausible biosynthetic pathways of depsidones, compounds **1**–**6**.

**Figure 7 marinedrugs-17-00266-f007:**
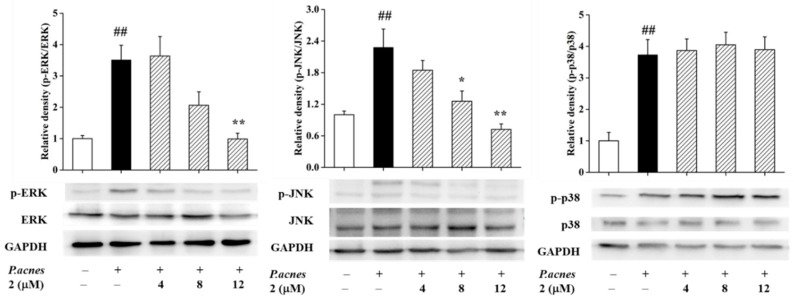
Impacts of **2** on the MAPK signaling pathway in *P. acnes*-induced THP-1 cells. The variations of ERK, JNK, and p38 proteins were detected by western blot. The data were exhibited as the mean ± SD of three independent experiments. *## p* < 0.01 compared to the control group; * *p* < 0.05 and ** *p* < 0.01 compared to only *P. acnes*-induced cells.

**Figure 8 marinedrugs-17-00266-f008:**
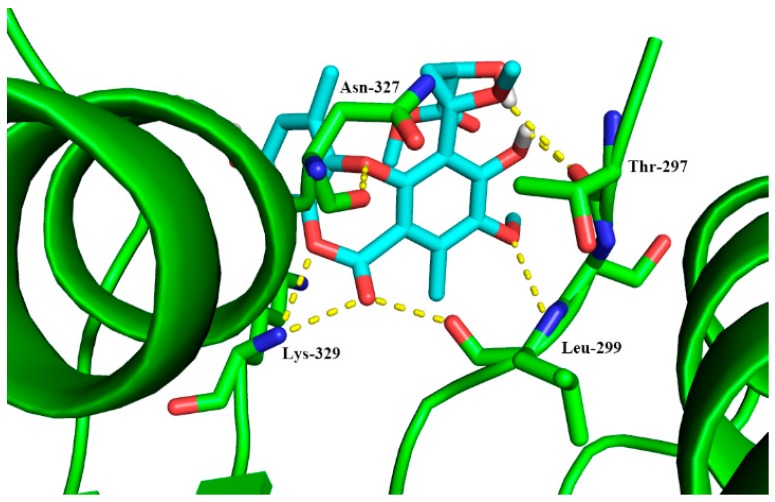
Docking analysis of compound **2** with TLR2/1 protein.

**Table 1 marinedrugs-17-00266-t001:** ^1^H (500 MHz) nuclear magnetic resonance (NMR) data of compounds **1**–**6** (δ in ppm, *J* in Hz) ^a^.

No.	1	2	3	4	5	6
3					6.65, s	
8	5.65, dd (11.1, 2.9)	5.65, dd (10.3, 3.8)	5.72, dd (11.1, 2.9)	5.74, dd (10.8, 3.5)	2.35, s	2.37, s
9a	1.93, ddd (14.0,10.5,2.9)	2.25, ddd (14.8, 5.3, 3.8)	1.92, ddd (14.0, 10.5, 2.9)	2.25, m		
9b	2.41, m	2.45, m	2.41, m	2.41, m		
10	4.53, dd (10.5, 2.7)	4.36, dd (5.5, 3.8)	4.54, dd (10.5, 2.7)	4.35, dd (5.5, 3.0)		
12	2.38, s	2.39, s	2.38, s	2.39, s		
3′	6.65, d (3.0)	6.63, d (3.0)	6.62, d (3.0)	6.61, d (3.0)	6.45, d (3.0)	6.59, d (3.0)
5′	6.47, d (3.0)	6.47, d (3.0)	6.46, d (3.0)	6.46, d (3.0)	6.47, d (3.0)	6.46, d (3.0)
7′	2.46, s	2.47, s	2.46, s	2.47, s	2.37, s	2.46, s
3-CH_2_OH						4.94, s
4-OCH_3_						4.01, s
5-OCH_3_	3.78, s	3.81, s	3.78, s	3.79, s	3.72, s	3.74, s
8-OCH_3_	3.54, s	3.40, s				
8-OCH_2_CH_3_			1.31, t (7.0)	1.20, t (7.0)		
			3.73, q (7.0)	3.59, q (7.0)		
11-OCH_3_	3.79, s	3.79, s	3.80, s	3.83, s		
4-OH	9.27, s	9.07, s	9.57, s	9.36, s		

^a^ The ^1^H NMR data of **5** were measured in CD_3_OD; others were measured in CDCl_3_.

**Table 2 marinedrugs-17-00266-t002:** ^13^C (125 MHz) NMR data of compounds **1**–**6** (δ in ppm) ^a^.

No.	1	2	3	4	5	6
1	113.6, C	113.7, C	113.6, C	113.7, C	113.9, C	118.3, C
2	156.2, C	156.2, C	155.9, C	155.8, C	160.6, C	156.1, C
3	113.8, C	113.7, C	114.5, C	114.3, C	106.6, C	123.5, C
4	154.2, C	154.2, C	154.5, C	154.3, C	156.5, C	155.8, C
5	144.5, C	144.6, C	144.8, C	144.8, C	145.6, C	148.7, C
6	137.2, C	137.3, C	137.1, C	137.1, C	137.6, CH	137.4, C
7	163.8, C	163.7, C	163.5, C	163.5, C	165.0, C	163.5, C
8	75.9, CH	75.4, CH	74.4, CH	73.5, CH	13.9, CH_3_	14.1, CH_3_
9	40.2, CH_2_	38.7, CH_2_	38.7, CH_2_	38.4, CH_2_		
10	67.3, CH	67.4, CH	67.3, CH	67.3, CH		
11	175.4, C	175.6, C	175.5, C	175.8, C		
12	14.4, CH_3_	14.4, CH_3_	14.4, CH_3_	14.4, CH_3_		
1′	143.4, C	143.5, C	143.6, C	143.7, C	143.6, C	143.6, C
2′	144.6, C	144.7, C	144.6, C	144.6, C	146.1, C	144.8, C
3′	106.0, CH	106.0, CH	106.1, CH	106.0, CH	105.8, CH	105.7, CH
4′	153.3, C	153.1, C	153.0, C	152.8, C	155.8, C	153.0, C
5′	114.5, CH	114.5, CH	114.4, CH	114.2, CH	114.7, CH	114.5, CH
6′	131.0, C	131.0, C	131.0, C	131.1, C	132.4, C	132.1, C
7′	18.2, CH_3_	18.1, CH_3_	18.2, CH_3_	18.0, CH_3_	16.0, CH_3_	17.0, CH_3_
3-CH_2_OH						55.3, CH_2_
4-OCH_3_						61.2, CH_3_
5-OCH_3_	52.9, CH_3_	52.8, CH_3_	52.9, CH_3_	52.7, CH_3_	60.7, CH_3_	60.3, CH_3_
8-OCH_3_	58.1, CH_3_	57.9, CH_3_				
8-OCH_2_CH_3_			66.7, CH_2_	66.3, CH_2_		
			15.3, CH_3_	15.0, CH_3_		
11-OCH_3_	60.4, CH_3_	60.4, CH_3_	60.3, CH_3_	60.3, CH_3_		

^a^ The ^13^C NMR data of **5** were measured in CD_3_OD; others were measured in CDCl_3_.

**Table 3 marinedrugs-17-00266-t003:** Anti-inflammatory activities of partial compounds in *Propionibacterium acnes*-induced THP-1 cells.

Compound	Safe Concentration (µM) ^a^	IC_50_ (mean ± SD, *n* = 3, µM)
**1**	6.25	/ ^b^
**2**	12.5	7.47 ± 0.35
**5**	50	/
**6**	40	18.83 ± 0.65
**Retinoic acid ^c^**		3.38 ± 0.28

^a^ Ensuring the compounds were not cytotoxic to the THP-1 cells. ^b^ The IC_50_ was not under the safe concentration. ^c^ Positive control.

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
