# Peer review of "Curdepsidones B–G, Six Depsidones with Anti-Inflammatory Activities from the Marine-Derived Fungus Curvularia sp. IFB-Z10"

_marinedrugs, 2019, doi:10.3390/md17050266_

Reviewer 1 Report

The authors reports six new analogs from a marine-derived fungus and their anti-inflammatory activities. The structures of six new compounds were elucidated based on spectroscopic data analyses in combination with computer assisted calculation. It is an interesting article to the readers of Marine Drugs. However, there are several unclear parts to be clarified. Therefore, these issues should be considered before acceptance.

Comments:

1. It seems that the original compound, curdepsidone A was reported by the authors. It would be great if the biological activity of original compound are included in the manuscript. If any known activity of curdepsidone A, it should be included in the introduction of the manuscript. It is very critical aspects related to the novelty of research.

2. Structure elucidation was accomplished by spectroscopic data analysis and computer assisted calculation. However, there are several way to confirm the position of OMe and OH on the aromatic rings.

- For OMe position, we can use NOESY or ROESY spectra analysis. Or long-range coupling constant modified HMBC can give insights,

- Position of OH on aromatic ring can be determined by acetylation.

3. Several key HMBC correlations for the security of the structure are missing. A long-range coupling constant modified HMBC can show 4-5 bond H-C couplings on aromatic rings and it can secure the position of substitution on aromatic rings.

4. The exact mass (m/z) values should be calculated under the consideration of charge state. If the ion is negatively charged, m/z values should be calculated from [M-H]- rather than [M-H].

5. IC50 values in the table 3 and text are inconsistent. Please re-check the data. Also, replication for statistical analyses were missing. That should be described in the manuscript.

6. In table 3, a brief explanation for the symbol of “/” needs to be added in the footnote.

7. Experimental detail on ECD measurement should be added in the manuscript. In addition, is there any reason to select gaseous phase for the calculation of ECD spectra?

8. The biosynthesis of these compounds would be very interesting. Therefore, please include a paragraph on the biosynthetic speculation. Also, add the biosynthetic speculation on the epimeric pairs of compounds at C-8.

Line 98: unites è units

Lines 276 and 281: the number in “CO2” needs to be written in subscript.

The NMR solvent in the manuscript and the supplementary metarials are not consistent.

Author Response

Response:

    Thanks so much for your kind comments on Curdepsidones B-G, Six Depsidones with Anti-inflammatory Activities from the Marine-Derived Fungus Curvularia sp. IFB-Z10. And we provided a point-by-point response to your comments in the Word file.

Reviewer 2 Report

Depsidones are diaryl cyclic compounds linked by an ether and ester linkages joining the two aromatic rings. This class of compounds are interesting because of their unique chemical structures and bioactivities. This is a well written manuscript that describes the isolation and characterization of 6 new depsidones. The compound structures and stereochemistry are established with extensive NMR spectroscopic studies and CD spectra. In addition, one of the compounds, Curdepsidone C (2) was found to display significant anti-inflammatory activity and reduced the P. acnes induced phosphorylation levels of JNK and ERK in a dose-dependent manner. Overall, this is an excellent manuscript that fits in to the mission of Marine Drugs. I recommend its publication.

Author Response

Response:

    Thanks so much for your kind comments on the Curdepsidones B-G, Six Depsidones with Anti-inflammatory Activities from the Marine-Derived Fungus Curvularia sp. IFB-Z10.

Reviewer 3 Report

This review concerns the manuscript, Curdepsidones B-G, Six Depsidones with Anti-inflammatory Activities from the Marine-Derived Fungus Curvularia sp. IFB-Z10, Journal: Marine Drugs, Manuscript ID: marinedrugs-498495.

My opinion can be valid for structure determination.  The presented six structures of the compounds depsidones were correctly determined using NMR spectroscopy. The information presented in Tables 1 and 2 correspond to the spectra included in Supporting Information. The spectra are of good quality, readable, the position of all signals are marked, integration is given in 1H NMR data. In each spectrum the structure of compound is placed, which makes the interpretation comfortable. Also, 2D NMR spectra are clearly presented, which is not common in submitted manuscripts. One small remark: the structure presented in Figure 1 should be a little more bigger, just as in Figures 2 and 3. The manuscript should be accepted.

Author Response

Response:

     Thanks so much for your kind comments on the Curdepsidones B-G, Six Depsidones with Anti-inflammatory Activities from the Marine-Derived Fungus Curvularia sp. IFB-Z10. And we have provided a point-by-point response to your comments in the Word file.

Round  2

Reviewer 1 Report

The authors revised manuscript as the reviewers suggested. Therefore, it would be acceptable to be published in Mar Drugs.